# Long-term mammal herbivory on arthropod assemblages at Kruger National Park, South Africa

**Ludzula Mukwevho**[1,2]*, **Tatenda Dalu**[2], **Frank Chidawanyika**[1,3]

**1** Department of Zoology and Entomology, University of the Free State, Bloemfontein, South Africa, **2** School of Biology and Environmental Sciences, University of Mpumalanga, Mbombela, South Africa, **3** International Center of Insect Physiology and Ecology (ICIPE), Nairobi, Kenya

* ludzula.mukwevho@ump.ac.za

**Data Availability Statement:** The data file collated for this study is available online in the figshare repository: https://doi.org/10.6084/m9.figshare.22663963.

## Abstract

Protected savannas are essential reserves for biological diversity, including endangered arthropod species, however, extreme grazing by mammals has cascading impacts on the communities and disrupts the functioning of these ecosystems globally. The current study assessed the abundance, species richness and composition of arthropods at the long-term grazing exclosures of Kruger National Park, South Africa. Pitfall traps and active searches were used to sample arthropods at the ungrazed, moderately, and heavily grazed exclosures. We found that Hymenoptera, Coleoptera, Diptera and Araneae were the most abundant orders of arthropods sampled. The abundance of multi-taxon, Diptera and Hymenoptera was significantly different between exclosures. In contrast, Coleoptera had high numbers of morphospecies compared to Hymenoptera, Araneae and Diptera. Species richness for multi-taxon, Diptera and Hymenoptera was significantly high at the heavily grazed compared to moderately grazed and ungrazed exclosures. Up to 22.2%, 41.2%, and 44.4% of the morphospecies were unique to the ungrazed, moderately and heavily grazed exclosures, respectively. A high proportion of morphospecies shared between exclosures were Coleoptera (41.0%) and Hymenoptera (38.5%) compared to Diptera and Araneae with less than 5% recorded across exclosures. Thus, morphospecies within the least abundant orders, namely Diptera and Araneae, were represented by singletons that were unique to particular exclosures compared to the most abundant arthropod orders (e.g., Coleoptera and Hymenoptera). We conclude that long-term mammal grazing enhances species richness and niche composition together with sparse and unique arthropods in the protected savannas. Therefore, managed grazing regimes can serve as a tool for maintaining the integrity of the protected savannas.

## Introduction

The magnitude of impact posed by herbivorous mammals depends on their size, grazing and browsing intensities coupled with their stocking rates [1,2]. Grazing intensities (top-down

**Funding:** The study was funded by the University of Mpumalanga, Institutional Research Theme (IRT: Biodiversity Conservation and Development), the University of Free State (119DD2789) and NRF 119N4588. The funders had no role in study design, data collection and analysis, the decision to publish, or preparation of the manuscript.

**Competing interests:** The authors have declared that no competing interests exist.

controls/ drivers) have incandescent effects on the progression, heterogeneity and diversity of primary ecological producers (i.e., plants) and therefore yields both direct and indirect influence on the abundance and diversity of arthropods at different trophic cascades [3–8]. Immense and continuous pressures threatens the abundance of highly sensitive morphospecies of arthropods, whereas extreme impacts may lead to extinction [9,10].

The reduction or elimination of arthropods in the savannas, grasslands and forests disrupts specific functions such as pollination, plant nipping and decomposition of organic materials in the ecosystem [2,7,10,11]. For example, the disruptions in the ecological processes at different trophic cascades can indirectly influence the abundance and diversity of coprophages and detritivores [11–13]. Thus, incorporation of organic materials (e.g., litter, carcass and plant materials) into the soil is prolonged [13]. In turn, minimal decomposition due to the limited activity of coprophages and detritivores compromises soil fertility and vegetation structural complexicity which subsequently reduces the grazing capacity [14,15]. Therefore, an indepth understanding of arthropod assemblages in relation to grazing may aid conservation decisions by managers of protected areas and subsequent policy reforms [16,17].

Kruger national park (KNP) is amongst the largest protected area in sub-Saharan Africa and is ranked seventh in the African continent [18]. The park was officially proclaimed as protected in 1898 with the intention of conserving mammals whose abundance and richness was threatened by anthropogenic activities and climate change [19–21]. Gradual decline in the number of grazers per unit area was attributed to poaching, predation, disease, climate change and culling during seasons where food resource is inadequate at the protected nature reserve [22–25]. Change in the number of grazers influences heterogeneity of landscapes, abundance, diversity and composition of plants, mammals, birds and arthropods [1,26–28]. Thus, the top-down pressures inflicted by grazers should be thoroughly understood to encourage conservation of species in the ecosystem. Well managed protected savanna encourages optimal functioning of the protected savannas, whilst, restricted anthropogenic activities enhances the conservation of plants, birds, and arthropods in the protected area [29–32].

Grazing by stock and game mammals is common across savanna and grassland biomes of rangeland and conservation importance [2,4,33]. Numerous studies previously assessed the spatial and temporal response of plants, mammals, arthropods, and soil biota to grazing, but the long-term impact of grazing is poorly understood [34–36]. The current study sought to evaluate the sensitivity of multi-taxon and the most abundant yet under-investigated groups of biological organisms (i.e., arthropods) to different grazing intensities at the long-term exclosures of KNP. Here, we quantified the long-term impact (i.e., direct and indirect) of grazing on arthropods inhabiting the protected savanna biome of KNP. We hypothesized that grazing negatively affects the abundance of arthropods, but, encourages conservation of species and diversity of arthropods at KNP.

## Materials and methods

### Ethics statement

The collection of arthropods to measure the impact of grazing mammals on the diversity of arthropods at the protected savanna was done after the permit (UFS-AED2018/0078) was ethical clearance issued on the 21st of February 2019 by the Interfaculty Animal Ethics Committee of the University of Free State. A research permit (MUKWL1570) allowing access and collection of specimens at a protected area was issued through the Scientific Services division of Kruger National Park. The study was declared ethically compliant and ecologically safe by these organizations. Mammals were not collected or harmed during this trial, and no endangered or protected species were at risk.

## Herbivore exclosures at Kruger National Park

Long-term mammal herbivory exclosures were established in 2002 at the Nkuhlu (24˚ 59.168'S; 31˚ 46.540'E) and Letaba regions (23˚ 45.267'S; 31˚ 25.916'E) to determine the spatial and temporal changes in the heterogeneity of vegetation following different mammal herbivory and fire regimes at the protected KNP [37,38]. Approximately 50 ha exclosures were demarcated and fenced (i.e., moderately and fully hereafter referred to as moderately grazed and ungrazed exclosures, respectively) along the Sabie and Letaba rivers were used in the current study.

Of the mammal grazing exclosures, both megaherbivores and mesoherbivores were excluded at the ungrazed exclosure, but only megaherbivores were excluded at the moderately grazed exclosure. Ungrazed exclosure was entirely electrified, and the diamond mesh wire (1.2 m height; 63 x 2.5 mm pores) was erected to restrict all grazers (e.g., small mammals, meso- and mega-herbivores) and browsers from accessing the exclosure. A concrete lintel (i.e., 200 x 100 mm) was also erected to prevent rhinarium digging grazers such as warthog from entering the ungrazed exclosure. At a moderately grazed exclosure, two electrified wires were erected at a height of 1.8 m and 2.2 m to exclude megaherbivores [e.g., Elephants, *Loxodonta Africana* (Proboscidea: Elephantidae), Rhinoceros, *Diceros* sp. / *Ceratotherium* sp. (Perissodactyla: Rhinocerotidae) and Giraffe, *Giraffa camelopardalis* (Artiodactyla: Giraffidae)]. Lastly, ungrazed and moderately grazed exclosures were separated by an open field (Nkuhlu: 25 ha, Letaba: 36 ha) (i.e., hereafter referred as heavily grazed exclosure) which was accessible to all sizes of herbivorous mammals (e.g., meso- and megaherbivores) and the plot was used as a control [37–39] (Fig 1).

## Experimental site descriptions

Nkuhlu and Letaba exclosures were established along the southern and northern periphery of KNP with sandy (granite) soils. The tropical arid region has an average temperature of 26.3 ºC and summer rainfall of between 400 mm and 600 mm per annum. The Nkuhlu site is dominated by the dense woody vegetation mostly *Acacia nigrescens* and *Combretum apiculatum*, whilst the Letaba site is dominated by the rugged veld mainly constituted of *Mopane/ Combretum*.

## Arthropod sampling

Arthropods were collected using pitfall traps and active searches during the Austral winter at the KNP exclosure plots. Within each exclosure (i.e., ungrazed, moderately and heavily grazed), five pairs of pitfalls were temporarily placed at 20 m intervals within a 100 m transect, and the experiment was replicated three times. A 500 mL (i.e., 8 cm diameter and 10 cm height) plastic honey jars were buried with rims flushing the soil surface and half-filled with trapping fluid (e.g., 1:1 proportion of ethyl glycol and water). Pitfall traps remained open for five consecutive days during arthropod sampling and thereafter, the containers were removed with holes being filled-up with soil thereafter to allow soil recovery. Active searches were conducted within the transect for 30 minutes to improve arthropod-capturing efforts, and samples were pooled with pitfalls. Contaminants such as mud and plant materials were washed, and specimens were transferred to 70% ethanol for preservation at the University of Mpumalanga laboratory. Specimens were sorted and identified to the morphospecies level (S1 Table). Identified voucher specimens and collections are housed at the University of Mpumalanga entomology laboratory.

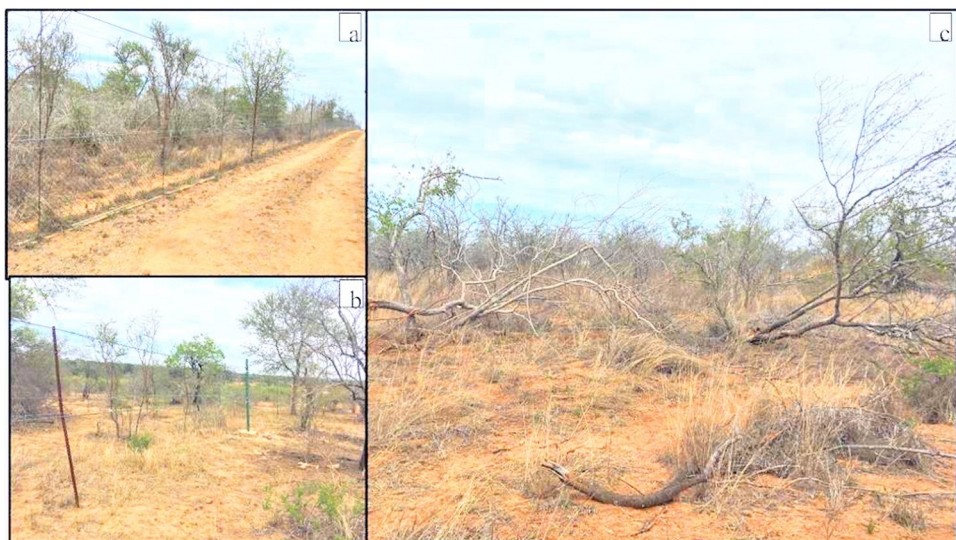

**Fig 1. Physical features of the demarcated mammal herbivory experimental plots at Kruger National Park.**
Ungrazed (a) which is not accessible by the herbivorous mammals; and moderately grazed (b) accessible to mesoherbivores; separated by the heavily grazed (open field) (c) which is accessible to all grazing mammals (e.g., mega and mesoherbivores).

## Statistical analysis

Data analyses were performed using Plymouth Routines In Multivariate Ecological Research (PRIMER-e), and STATISTICA version 13.3 (TIBCO Statistica$^{TM}$). EstimateS version 9.1.0 (R. K. Colwell) was used to predict the asymptotic arthropod species richness for multi-taxon and each of the four abundant arthropod orders (i.e., Araneae, Coleoptera, Diptera and Hymenoptera (Formicidae). The least-biased estimators, namely Chao1, Chao2, Jacknife1, Jacknife2 and Michaelis–Menten mean estimators were used to measure the adequacy of sampling effort [40,41]. Samples were randomised 100 times.

Shapiro–Wilks test was used to assess the normality of the abundance and species richness data for multi-taxon and within individual orders; Araneae, Coleoptera, Diptera and Hymenoptera (Formicidae). The normally distributed data was analysed using One-way analysis of variance (ANOVA) to measure the statistical differences in abundance and species richness of multi-taxon, Araneae, Coleoptera, Diptera and Hymenoptera sampled at the ungrazed, moderately and heavily grazed exclosures of KNP. Tukey's HSD (Honestly Significant Difference) *post-hoc* test was then used to determine the differences in abundance and species richness of arthropods sampled at the three exclosures.

Non-metric multidimensional scaling (nMDS) in PRIMER was used to detect the trends of dissimilarities of species composition between ungrazed, moderately and heavily grazed exclosures of KNP. The assemblage of multi-taxon, Araneae, Coleoptera, Diptera and Hymenoptera were determined using Bray-Curtis indices and the number of species unique or those shared between ungrazed, moderately and heavily grazed exclosures were presented on Venn diagrams.

## Results

A total of 3 526 individual arthropods representing 8 orders, 29 families and 111 morphospecies were sampled at the ungrazed, moderately and heavily grazed exclosures at KNP. Of these, 1 676 individual arthropods were collected from the ungrazed exclosure, whereas 994 and 856

individuals were recorded at the moderately and heavily grazed exclosures, respectively. Hymenoptera and Coleoptera were the most dominant orders of arthropods, with 2 587 (73.4%) and 798 (22.6%) individuals comprising of 4 (Formicinae, Myrmicinae, Ponerinae and Pseudomyrmecinae) subfamilies of Formicidae and 6 (Carabidae, Curculionidae, Gyrinidae, Hydrophilidae, Staphylinidae and Tenebrionidae) families from 27 and 32 morphospecies, respectively. Furthermore, 37 (1.4%) and 32 (0.9%) individual arthropods were sampled from Diptera and Araneae orders comprising 9 and 17 morphospecies from 4 and 7 families, respectively. The abundance of individuals sampled from the remaining orders of arthropods (e.g., Blattodea, Hemiptera, Lepidoptera and Orthoptera) ranged between 12 (0.3%) and 23 (0.7%). For each order of arthropods, up to 8 morphospecies were recorded from at least 4 families (S1 Table).

The abundance of multi-taxon significantly varied ($F_{(2,15)}$ = 14.920, $P < 0.001$) between ungrazed, moderately and heavily grazed exclosures. Arthropods were most abundant at the ungrazed exclosure, with a noticeable decline of 1.7 and 1.9 folds at the moderately and heavily grazed exclosures, respectively. A separate analysis showed that the abundance of Hymenoptera and Diptera ($F_{(2,15)}$ = 3.710, $P = 0.049$) were significantly affected ($F_{(2,15)}$ = 10.579, $P < 0.001$) by grazing. The abundance of Hymenoptera was higher at the ungrazed exclosure, with a slightly extreme decline of 1.9 and 2.5 folds at the moderately and heavily grazed exclosures, respectively. Contrarily, the abundance of Diptera increased by up to 3.1 folds on the grazed compared to the ungrazed exclosures (Fig 2). The abundance of Araneae ($F_{(2,15)}$ = 0.109, $P = 0.898$), Coleoptera ($F_{(2,15)}$ = 0.499, $P = 0.617$) were not significantly different between exclosures. The abundance of Coleoptera was highest at the ungrazed compared to grazed exclosures, but, the abundance of Araneae was contrasting.

Refraction curves were calculated for multi-taxon and each of the four abundant orders of arthropods separately. The correlation between the number of morphospecies and sampling efforts showed that estimators were nearing asymptote proportions between the observed species richness (observed = 111) and Chao1 (130.29 ± 9.94), Chao2 (131.88 ± 10.04), Jacknife2 (152.83), MM (121.59) for multi-taxon. Similarly, the estimators were nearing asymptote for Araneae (observed = 17, Chao1 = 19.54 ± 2.75, Chao2 = 20.98 ± 3.87, Jacknife2 = 26.97, MM = 41.75), Coleoptera (observed = 32, Chao1 = 36.24 ± 5.37, Chao2 = 33.98 ± 3.17, Jacknife2 = 38.98, MM = 33.14), Diptera (observed = 9, Chao1 = 9.97 ± 1.83, Chao2 = 11.24 ± 3.38, Jacknife2 = 12.98, MM = 12.35), and Hymenoptera (observed = 27, Chao1 = 33 ± 7.28, Chao2 = 30.31 ± 4.11, Jacknife2 = 34.95, MM = 26.95) (Fig 3). Species accumulation showed that multi-taxon were 16.7% lower than the observed morphospecies, whilst up to 32.1%, 21.9%, 12.9% and 9.7% declines were observed for Araneae, Diptera, Coleoptera, and Hymenoptera, respectively.

The species richness of multi-taxon significantly differed ($F_{(2,15)}$ = 15.272, $P < 0.001$) between ungrazed, moderately and heavily grazed exclosures. At total of 47 morphospecies were recorded at the ungrazed exclosure. Noticeable exponential decline of 11.3% and 30.9% was recorded at the moderately and heavily grazed exclosures, respectively. Individual analysis for each order showed significant differences in the species richness within Diptera ($F_{(2,15)}$ = 6.759, $P = 0.008$) and Hymenoptera ($F_{(2,15)}$ = 6.696, $P = 0.008$), but not for Araneae ($F_{(2,15)}$ = 0.500, $P = 0.616$) and Coleoptera ($F_{(2,15)}$ = 3.346, $P = 0.063$). Species richness of multi-taxon, Diptera and Hymenoptera was not significantly different between ungrazed and moderately grazed exclosures but significantly higher at the heavily grazed exclosure (Fig 4). Irrespective of up to 4.7 and 1.3 folds increase in the species richness of Araneae and Coleoptera, there was no significant difference in the richness of species sampled at different experimental exclosures.

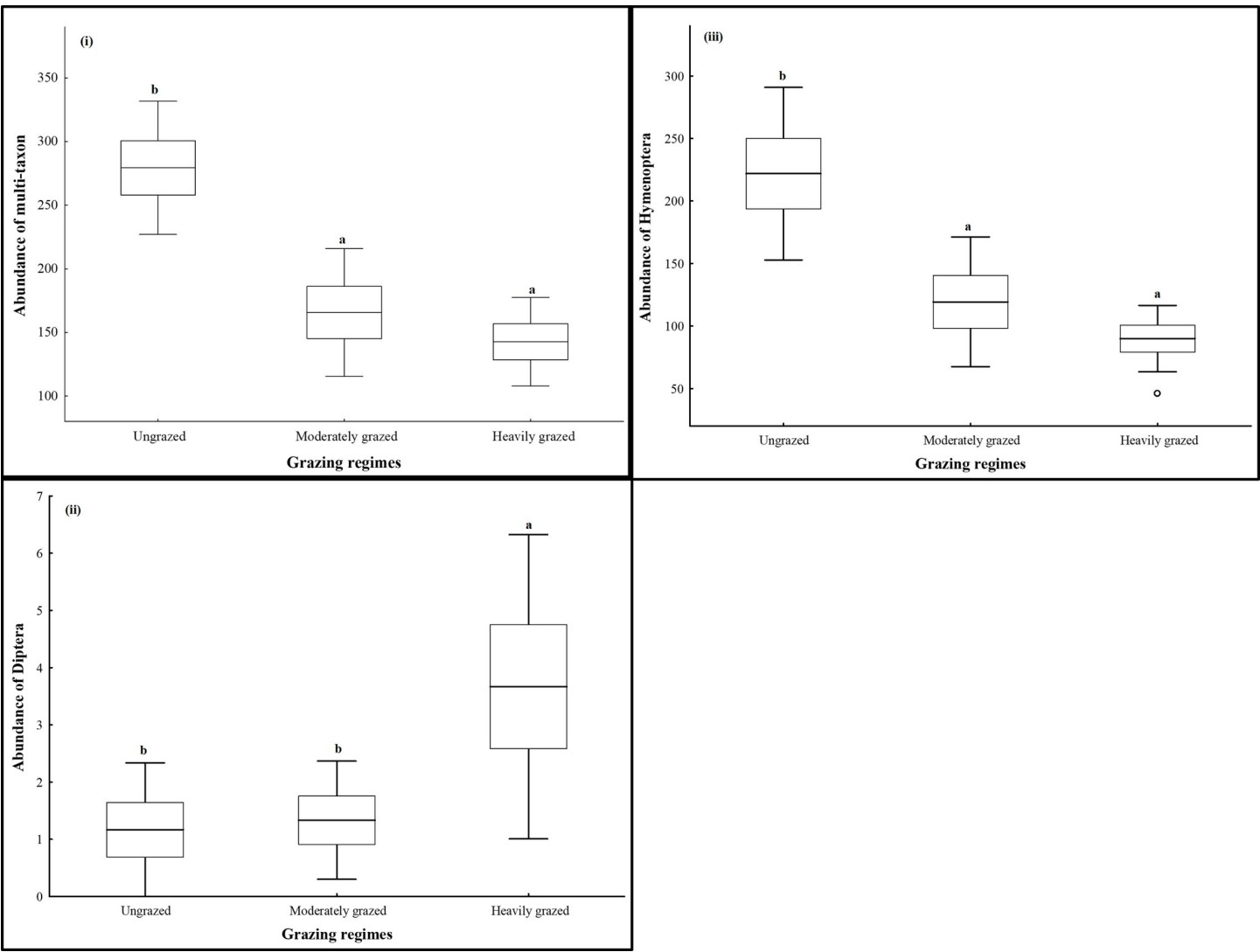

**Fig 2.** Boxplots showing the abundance of arthropods (i) and Hymenoptera (ii) sampled at the ungrazed, moderately and heavily grazed exclosures of Kruger National Park. Different letters above the bars show significant differences between treatments (Tukey HSD test: $P < 0.05$).

A total of 47 (42.9%) of the sampled morphospecies of arthropods were represented by singletons. Of the four most abundant orders of arthropods, Coleoptera had the least proportion of morphospecies which were recorded as singletons (28.1%) compared to Hymenoptera (33.3%), Diptera (44.4%), and Araneae (70.6%). Furthermore, 42.9%, 62.5% and 71.4% of morphospecies were sampled from the Orthoptera, Blattodea and Hemiptera orders of arthropods, respectively. From the order Lepidoptera, no singleton was recorded. The proportion of singletons was highest at the least abundant orders (i.e., Araneae, Blattodea, Diptera, Hemiptera and Orthoptera) compared to the orders where high numbers of arthropods (e.g., Coleoptera and Hymenoptera) were recorded (S1 Table).

Thirty-one of 111 morphospecies sampled at KNP were unique to the heavily grazed exclosure, hence, 24 and 17 were unique to the moderately grazed and ungrazed exclosures, respectively (Fig 5). Of the separate orders of arthropods, up to 22.2%, 41.2% and 44.4% of morphospecies were unique to the ungrazed, moderately and heavily grazed exclosures.

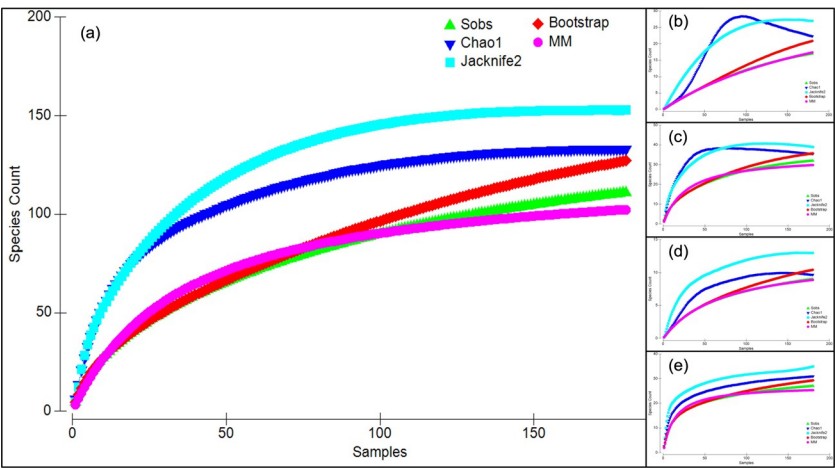

**Fig 3.** Morphospecies accumulation curves (mean of 100 randomizations) for all arthropods (a), Araneae (b), Coleoptera (c), Diptera (d), and Hymenoptera (e) sampled at the ungrazed, moderately and heavily grazed exclosures of Kruger National Park. The number of morphospecies observed (▲), Chao1 (▼), Jacknife2 (■), Bootstramp (◆) and MM (●) estimators are shown on the graphs.

Eighteen morphospecies were shared between 3 exclosures and of those, at least 10 were Hymenoptera, 7 were Coleoptera, and the last 1 was Diptera. A total of 10, 9 and 2 morphospecies were shared between ungrazed and heavily grazed, moderately and heavily grazed, and moderately grazed and ungrazed exclosures, respectively. At least 50% and 30% of morphospecies shared between ungrazed and heavily grazed exclosures were Coleoptera and Hymenoptera, whilst up to 44%, 33% and 22% of those shared between moderately and heavily grazed exclosures were others (i.e., less abundant morphospecies namely Blattodea, Hemiptera, Lepidoptera and Orthoptera), Coleoptera and Hymenoptera, respectively. Lastly, of the 2 morphospecies shared between ungrazed and moderately grazed exclosure, 1 was Coleoptera, hence, the remaining was the other (i.e., Blattodea) (Fig 5 and S2 Table). Bray-Curtis dissimilarity indices were more significant for Araneae, Diptera, and Other morphospecies compared to Coleoptera and Hymenoptera sampled at ungrazed, moderately and heavily grazed exclosures at KNP. Non-metric Multidimensional Scaling (nMDS) ordination plots ascertains that most species were shared between ungrazed, moderately and heavily grazed with least species appearing to be outliers for multi-taxon and each order separately (Fig 6).

## Discussion

In the current study, Araneae, Coleoptera, Diptera and Hymenoptera were amongst the most dominant orders of arthropods captured at the ungrazed, moderately and heavily grazed exclosures at the protected area of KNP. Of the few studies that assessed the impact of grazing on multi-taxon, Araneae, Coleoptera, Diptera and Hymenoptera were listed among the dominating orders of arthropods in the protected grasslands, steppe and savannas [8,10,17,42–44]. Although Lepidoptera and Orthoptera were recorded as the least abundant orders in the current study, these orders were recorded amongst the four most abundant orders in some studies conducted in the protected areas of Sweden [10], South Africa [42,45] and the United States of America [17].

The abundance of multi-taxon was significantly high at the ungrazed compared to moderately and heavily grazed exclosures. These findings were similar to those reported by [17,42,46]. Irrespective of the notable similarity in the results between studies conducted at

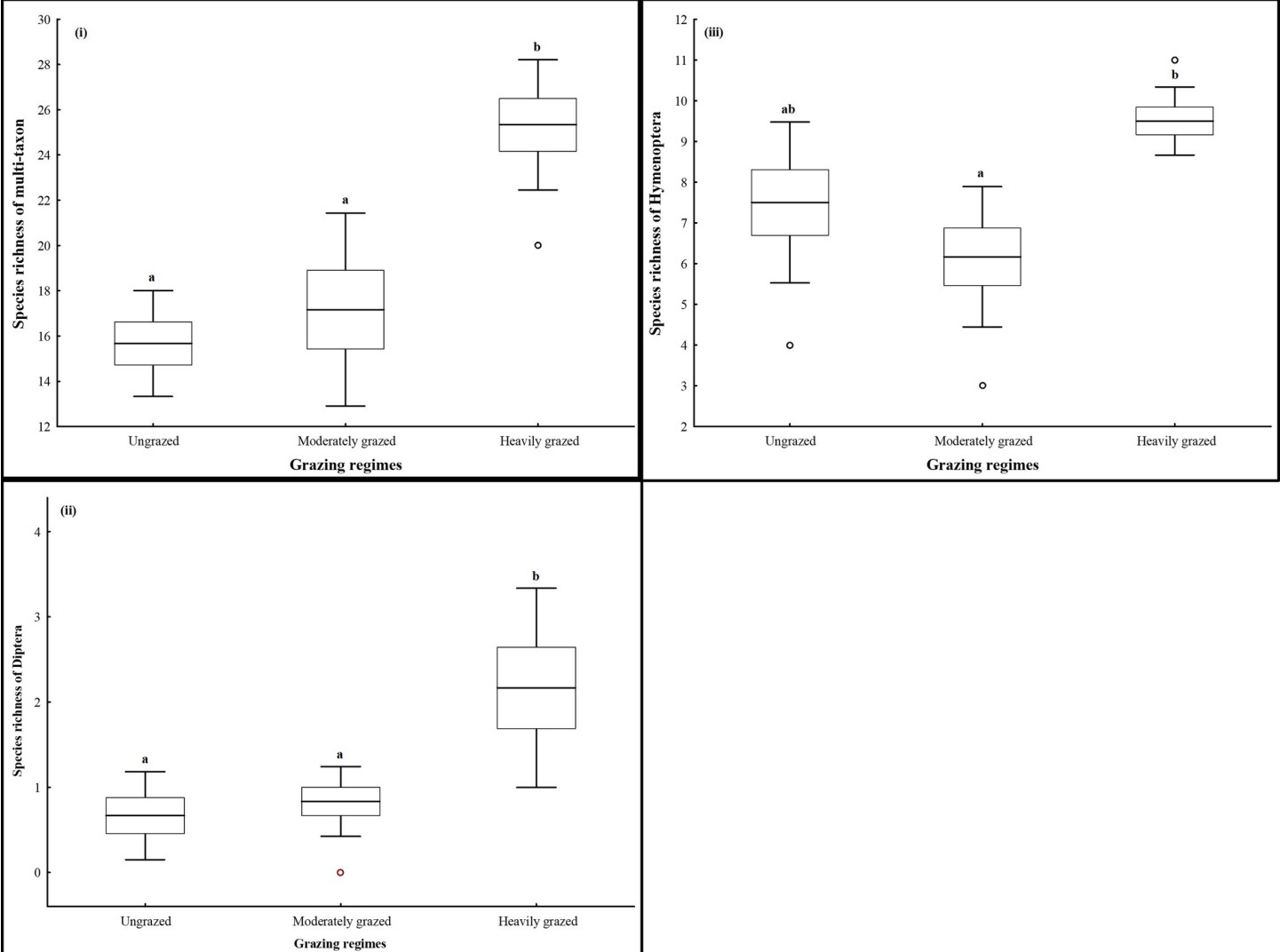

**Fig 4.** Boxplots showing the Species richness of arthropods (i), Diptera (ii) and Hymenoptera (iii) sampled at the ungrazed, moderately and heavily grazed exclosures of Kruger National Park. Different letters above the bars show significant differences between treatments (Tukey HSD test: $P < 0.05$).

KNP, the abundance multi-taxon collected in the current study (i.e., 17 years after the establishment of exclosures) was high compared to that of a study conducted at least 6 years after establishment of exclosures [5,42]. Irrespective of the significant differences in the abundance of Hymenoptera and Diptera in the current study, previous studies confirmed the significant sensitivity of Hymenoptera [5,47], not Diptera [2,17]. Furthermore, some studies reported contrasting results that demonstrated significant differences in the abundance of Araneae and Coleoptera when analysed separately [5,28].

The current study demonstrated significant difference on the species richness of multi-taxon, Diptera and Hymenoptera, hence, the richness of Araneae and Coleoptera did not vary between exclosures. Although the results of the current study corroborates with that of previous ones on the species richness of multi-taxon [45], Araneae [42,48], Coleoptera [17], and Hymenoptera [17,47], contary results showed that species richness of neither multi-taxon, Diptera nor Hymenoptera significantly differed between grazed and ungrazed exclosures at

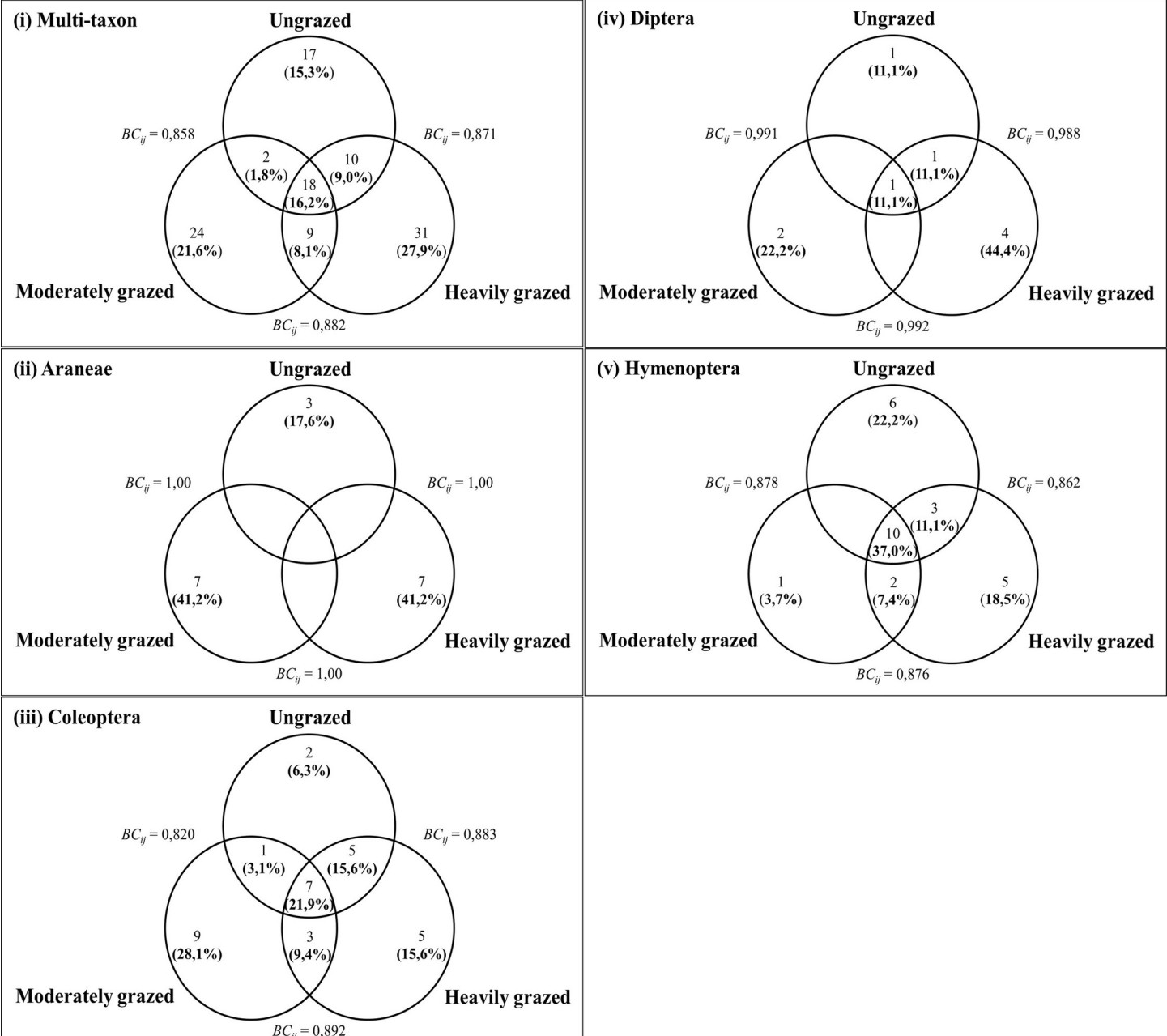

**Fig 5.** Venn diagram showing the number of morphospecies and percentages (in parentheses) for all arthropods (i), Araneae (ii), Coleoptera (iii), Diptera (iv) and Hymenoptera collected exclusively at the ungrazed, moderately and heavily grazed exclosures. Species share between exclosures and dissimilarity indices of Bray–Curtis (i.e., $BC_{ij}$).

the grassland and savanna landscapes [17,42]. Furthermore, species richness of Araneae and Coleoptera was reported to be highly sensitive to grazing [30,42,45,48,49], hence, the species richness was not significant between treatment exclosures in the current study.

Species accumulation metrices showed that the number of species sampled in the current study failed to reach an asymptote, and this is a common phenomenon in ecological studies where arthropods are used as ecological indicators [50–53]. Irrespective of the overall failure,

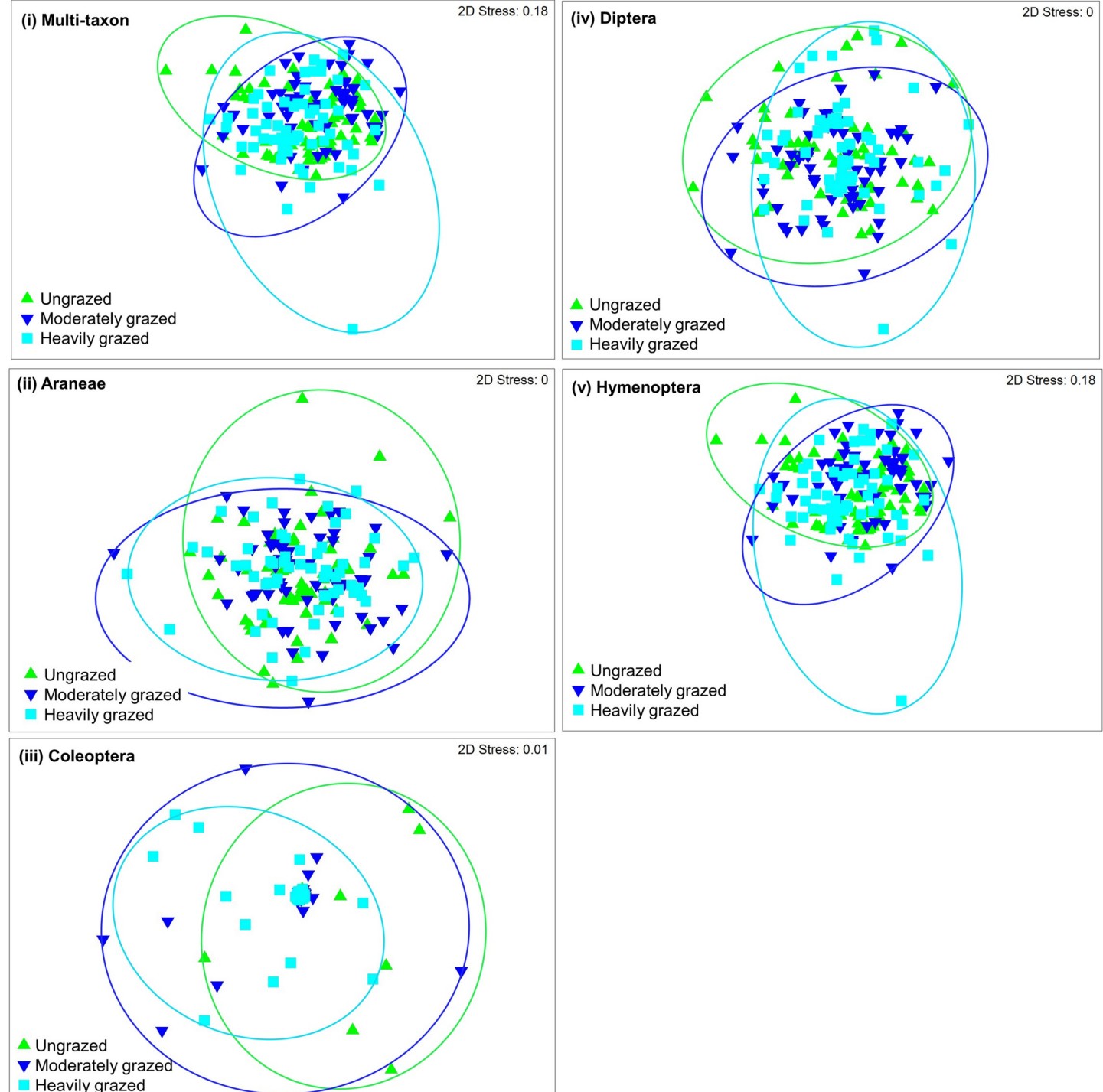

**Fig 6.** Non-metric Multidimensional Scaling (nMDS) ordination plots showing the composition of all species of arthropods (i) and separate orders of species such as Araneae (ii), Coleoptera (iii), Diptera (iv), and Hymenoptera sampled at different grazing regimes of the Kruger National Park.

orders (i.e., Hymenoptera and Coleoptera) with considerably lower numbers (i.e., ± 30%) of morphospecies represented by singletons were nearing asymptote, hence, those (i.e., Araneae and Diptera) with higher number (i.e., 42.8%) of singletons were distant from the point of

asymptote. Proportions of singletons reported in the current study corroborate with those reported from studies where sampling was intensive [53–55].

Inconsistency in the abundance, species richness, and rarerity of species collected at KNP and other protected savanna biomes could be attributed to sampling efforts, period and technique (i.e., sole or integrated) are amongst useful factors associated with adequacy in the collection of individual arthropods and morphospecies richness [41,56,57]. Nevertheless, increased sampling efforts and continuous sampling failed to improve the rarerity of arthropods were significantly higher compared to the current study [53–55,57–59]. Thus, continuous sampling using similar technique(s) neither improves the species richness nor warrants the possibility of sampled species reaching an asymptote [53,55]. Increased sampling efforts increases the abundance of arthropods and improve assemblage composition, but some new unique morphospecies (i.e., singletons) are sampled in the process [59,60]. Thus, continuous sampling in the current study may encourage increase in numbers of arthropods, but not reduce rarerity of morphospecies sampled at different treatment exclosures.

Irrespective of a perception which claims that the type of artthopod sampling technique(s) adopted optimizes arthropod capturing [61,62], pitfall trapping integrated with active searches optimized sampling of multi-taxon and separate orders of arthropods. The optimal efficiency of pitfall and active searches was demonstrated by high number of multi-taxon, Araneae, Coleoptera Diptera and Hymenoptera sampled at KNP in 2019, compared to those collected in a previous study conducted at the same region [5,42]. Integration of a different technique would optimize collection of specialist order of arthropods as demonstrated by the efficient sampling of Orthoptera in a study where sweep net was used to collect arthropods at KNP [5,42]. Furthermore, a different integration (e.g., pitfall trapping with active search, D-vac suction, sweep netting, malaise or light trapping) could improve sampling efforts of multi-taxon and separate orders of arthropods [2,17,28,63].

Since climate and sampling time (i.e., season) does not affect the abundance and richness of arthropods sampled at the savanna and grasslands of sub-Saharan Africa [64,65], plant diversity, stocking rates and grazing intensities could be associated with the varying response of arthropods at different exclosures of KNP [8,10,23,26,33,46,47,65–67]. Therefore, mammal activity and heterogeneity of plants needs to be understood and correlated with assemblages of arthropods to ascertain the direct and indirect impact of mammal grazing on the abundance, species richness and composition of arthropods at KNP.

## Conclusion and recommendations

Results of the current study show that grazing encourages conservation of arthropod species inhabiting the protected KNP although incandescent effects on the abundance was observed. Therefore, managed grazing in the KNP can be an important tool for the overall conservation of arthropods and to enhance heterogeneity of ecologically important insects and subsequent ecosystem function. In our study, we could not account for other abiotic factors such as rainfall and drought. Future studies may consider including these to account for their interactions with grazing regimes.

## Supporting information

**S1 Table. Numbers of morphospecies and their abundances for arthropods sampled at the ungrazed, moderately and heavily grazed exclosures of Kruger National Park.** Species classified to the order level were classified as "Others", but were distinguished by the unique morphological characteristics.
(DOCX)

**S2 Table. SIMPER analysis showing similarity of multi-taxon sampled at the ungrazed, moderately and heavily grazed exclosures of Kruger National Park.**
(DOCX)

## Acknowledgments

We thank Dr Salmina Mokgehle (University of Mpumalanga) for the constructive comments which improved the earlier draft of the manuscript. Dr Mduduzi Ndlovu (University of Mpumalanga) is thanked for his valuable inputs during conceptualization of the study. Special thanks to Sinenhlanhla Promise Mntambo (University of Mpumalanga) for the identification of the ants collected at KNP. Gratitude also goes to Ntombikayise Ndwandwe and the Entomology class of 2019 from the University of Mpumalanga for the field assistance and arthropod sorting. Lastly, we thank the editor and two anonymous reviewers for valuable suggestions and comments on the earlier version of the manuscript.

## Author Contributions

**Conceptualization:** Ludzula Mukwevho, Tatenda Dalu, Frank Chidawanyika.

**Data curation:** Ludzula Mukwevho, Frank Chidawanyika.

**Formal analysis:** Ludzula Mukwevho, Tatenda Dalu, Frank Chidawanyika.

**Funding acquisition:** Ludzula Mukwevho, Frank Chidawanyika.

**Investigation:** Ludzula Mukwevho, Tatenda Dalu, Frank Chidawanyika.

**Methodology:** Ludzula Mukwevho, Tatenda Dalu, Frank Chidawanyika.

**Project administration:** Ludzula Mukwevho, Frank Chidawanyika.

**Resources:** Ludzula Mukwevho, Frank Chidawanyika.

**Software:** Ludzula Mukwevho, Tatenda Dalu.

**Supervision:** Frank Chidawanyika.

**Validation:** Ludzula Mukwevho, Tatenda Dalu, Frank Chidawanyika.

**Visualization:** Ludzula Mukwevho, Tatenda Dalu, Frank Chidawanyika.

**Writing – original draft:** Ludzula Mukwevho, Tatenda Dalu, Frank Chidawanyika.

**Writing – review & editing:** Ludzula Mukwevho, Tatenda Dalu, Frank Chidawanyika.

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
