## [Decision Letter · Decision Letter 0]

17 Apr 2023

PONE-D-23-06410Long-term mammal herbivory on arthropod assemblages at Kruger National Park, South Africa.PLOS ONE

Dear Dr. Mukwevho, Thank you for submitting your manuscript to PLOS ONE. After careful consideration, we feel that it has merit but does not fully meet PLOS ONE’s publication criteria as it currently stands. Therefore, we invite you to submit a revised version of the manuscript that addresses the points raised during the review process. Please submit your revised manuscript by Jun 01 2023 11:59PM. If you will need more time than this to complete your revisions, please reply to this message or contact the journal office at plosone@plos.org. Please include the following items when submitting your revised manuscript:A rebuttal letter that responds to each point raised by the academic editor and reviewer(s). You should upload this letter as a separate file labeled 'Response to Reviewers'.A marked-up copy of your manuscript that highlights changes made to the original version. You should upload this as a separate file labeled 'Revised Manuscript with Track Changes'.An unmarked version of your revised paper without tracked changes. You should upload this as a separate file labeled 'Manuscript'.

We look forward to receiving your revised manuscript.

Kind regards,

Tunira Bhadauria, Ph.D.

Academic Editor

PLOS ONE

Journal Requirements:

"LM received funding from the University of Mpumalanga (Research Theme Funds) and FC funded the project through University of the Free State and National Research Foundation (NRF), South Africa. "

"We thank Dr Salmina Mokgehle (University of Mpumalanga) for the constructive comments which improved the earlier draft of the manuscript. Dr Mduduzi Ndlovu (University of Mpumalanga) is thanked for his valuable inputs during conceptualization of the study. Special thanks to Sinenhlanhla Promise Mntambo (University of Mpumalanga) for the identification of the ants collected at KNP. Gratitude also goes to Ntombikayise Ndwandwe and the Entomology class of 2019 from the University of Mpumalanga for the field assistance and arthropod sorting. We acknowledge funding from the University of Mpumalanga (Research Theme Funds) to LM and University of the Free State to FC who also received funding from the National Research Foundation (NRF), South Africa. "

"LM received funding from the University of Mpumalanga (Research Theme Funds) and FC funded the project through University of the Free State and National Research Foundation (NRF), South Africa. "

Reviewers' comments:

Reviewer's Responses to Questions

**Comments to the Author**

1. Is the manuscript technically sound, and do the data support the conclusions?

Reviewer #1: Partly

Reviewer #2: No

2. Has the statistical analysis been performed appropriately and rigorously? 

Reviewer #1: I Don't Know

Reviewer #2: Yes

3. Have the authors made all data underlying the findings in their manuscript fully available?

Reviewer #1: Yes

Reviewer #2: Yes

4. Is the manuscript presented in an intelligible fashion and written in standard English?

Reviewer #1: Yes

Reviewer #2: No

5. Review Comments to the Author

Reviewer #1: Comments for Authors:

1. Was a short-term observation period enough for finding out impacts of herbivory on arthropod populations? Were the migration patterns of herbivores taken into consideration?

2. How do the authors conclude that the arthropod richness variations are solely affected by grazing intensities and not by other biotic or abiotic factors? Physico-chemical properties of habitats, competition for resources, prey-predator relationships can also account for such variations in their abundances.

3. The discussion and conclusion sections are very weak. The site-specific differences in arthropod abundances need to be addressed adequately.

Reviewer #2: Dear Authors, after carefully reading the manuscript, it seems that it has lots of merit and suitable for publication in some less impact factor journal than Plos One. The major concerns that i have are 1. Data are not of much dimensions.

2. The study does not has much relevant and importance data with ecosystems, it does not mean that arthropods are not important part of the ecosystem.

3. If the study may include some more angles related with arthropods that involves some laboratory work also, then the MS could have been considered in this journal.

4. There are several sentences readability and grammar problems. 5. As per impact factor of the journal, the study is not enough to be accepted at this form of the MS.

6. PLOS authors have the option to publish the peer review history of their article (what does this mean?). If published, this will include your full peer review and any attached files.

Reviewer #1: No

Reviewer #2: No

<quillbot-extension-portal></quillbot-extension-portal>

---

## [Author Response · Author response to Decision Letter 0]

25 Apr 2023

Editor#

1. Ensure that your manuscript meets PLOS ONE's style requirements, including those for file naming.

Response: Authors formatted the manuscript according to PLOS ONE’s style and the attached files were also named according to the journal’s requirements.

2. We note that the grant information you provided in the 'Funding Information' and 'Financial Disclosure' sections do not match. When you resubmit, please ensure that you provide the correct grant numbers for the awards you received for your study in the 'Funding Information' section.

Response: The grant information was corrected and the research theme category was outlined as a reference (i.e., the funder does not assign a unique number). Furthermore, the University of Free State and NRF grant number was given. 

Please state what role the funders took in the study. If the funders had no role, please state: "The funders had no role in study design, data collection and analysis, decision to publish, or preparation of the manuscript." Please include this amended Role of Funder statement in your cover letter; we will change the online submission form on your behalf.

Response: Authors have declared that funders had no role in the research in the cover letter.

3. We note that you have provided funding information that is not currently declared in your Funding Statement. However, funding information should not appear in the Acknowledgments section or other areas of your manuscript. Please remove any funding -related text from the manuscript and let us know how you would like to update your Funding Statement. Please include your amended statements within your cover letter; we will change the online submission form on your behalf.

Response: The funding information was deleted in the Acknowledgments section and does not appear anywhere on the revised version of the manuscript. The funding statement was also revised and shared on the cover letter.

4. In your Data Availability statement, you have not specified where the minimal data set underlying the results described in your manuscript can be found. "Upon re-submitting your revised manuscript, please upload your study's minimal underlying data set as either Supporting Information files or to a stable, public repository and include the relevant URLs, DOis, or accession numbers within your revised cover letter.

Response: The data file collated for this study is available online in the figshare repository: https://doi.org/10.6084/m9.figshare.22663963. This information was presented on the cover letter for editor’s attention.

5. Please include your full ethics statement in the 'Methods' section of your manuscript file. In your statement, please include the full name of the IRB or ethics committee who approved or waived your study, as well as whether or not you obtained informed written or verbal consent. If consent was waived for your study, please include this information in your statement as well.

Response: The ethics statement was presented in the methods section. Names of committees, reference numbers of permits and the issue dates were outlined in this section. Ethical information presented here corresponds (i.e., although paraphrased) with that presented in the “Ethics statement”. 

Response to reviewer’s comments

Reviewer #1

1. Was a short-term observation period enough for finding out impacts of herbivory on arthropod populations? Were the migration patterns of herbivores taken into consideration?

Response: We strongly believe that the period of the study was sufficient and the data collected clearly elaborated variations in abundance, richness and composition of arthropods between grazing exclosures. As for migration patterns, our study design was based on the management regimes in the form exclosures implemented by the Park authorities. This therefore gives a controlled design were grazing intensity, which in this case could be deemed as a proxy for migration, was accounted for. In an open grazing plan, indeed accounting for direct migration, which would be dependent on availability of resources, would be very important. Thus, these long term grazing exclosures, enabled us to focus more on our major subject of interest in the form of arthropod responses. 

2. How do the authors conclude that the arthropod richness variations are solely affected by grazing intensities and not by other biotic or abiotic factors? Physico-chemical properties of habitats, competition for resources, prey-predator relationships can also account for such variations in their abundances.

Response: The role of different biotic and abiotic factors cannot be disputed in any field trial. However, all the plots were assessed under similarly prevailing conditions such that we could attribute the differences in arthropod responses to the management regimes. We have however expressed caveats in our conclusion and indicated that other factors may be investigated in future studies.

3. The discussion and conclusion sections are very weak. The site-specific differences in arthropod abundances need to be addressed adequately.

Response: The scope of the study did not include the site-specific factors outlined by the reviewer based on the assumptions given above and the inclusion of the suggested component in the discussion may therefore not add value to the current research. 

 

Reviewer #2: 

1. Dear Authors, after carefully reading the manuscript, it seems that it has lots of merit and suitable for publication in some less impact factor journal than Plos One. The major concerns that i have are 1. Data are not of much dimensions.

Response: Authors acknowledges complements on the merits of the manuscripts and still think PLOSONE is a suitable home for the manuscript. More importantly, we strongly feel that acceptance of a paper for publication should be solely based on the scientific merits of the study. Focusing on the journal metrices including IF is certainly subjective and prone to reviewer non-scientific elitist criteria for rejection e.g. geographic location or mere names of the authors as a basis for rejection. We would rather therefore be more drawn to discuss the science and not journal IF. As for dimensions, it is not clear whether the reviewer was referring to the mere measurable aspects of the study or its impact. Hence, for the comment to be constructive, it would have been highly appreciated if the reviewer had been elaborate citing the particular flaws or limitations of the study, which we could duly acknowledge if explicitly stated.

2. The study does not has much relevant and importance data with ecosystems, it does not mean that arthropods are not important part of the ecosystem.

Response: In our region, National Parks are a major asset as indicated in the manuscript and organismal interactions therein with management practices to track potential pitfalls for biodiversity conservation is of high importance. This motivation is clearly indicated in our motivation for the study in the manuscript. We therefore find this reviewer comment highly uniformed since biodiversity conservation is highly topical and a matter of societal concern. Moreover, the targeted study organisms provide key ecosystem services (and potentially disservices) that warrant attention. As discussed above, it would be highly helpful to get the actual flaws of the study than blanket discounting statements to enrich the scientific discourse and advancement of the science. 

3. If the study may include some more angles related with arthropods that involves some laboratory work also, then the MS could have been considered in this journal.

Response: Again, we find this comment highly speculative giving little to scientific discourse as lab work is not just done for the sake of it. What particular “angles related with arthropods that involves some laboratory work” would have sufficed? Was the laboratory work we conducted on species IDs enough? Without suggestions of particular study aspects that required laboratory investigations, we find the comment inconsequential. 

4. There are several sentences readability and grammar problems. 

Response: The manuscript was read several times including senior authors here who have no history of language problems. We strongly believe that the level of language use here is of acceptable level.

5. As per impact factor of the journal, the study is not enough to be accepted at this form of the MS.

Response: The authors respect the view of the reviewer on the suitability of the manuscript and also appreciate the constructive inputs given.

---

## [Editor Report · Decision Letter 1]

15 May 2023

Long-term mammal herbivory on arthropod assemblages at Kruger National Park, South Africa.

PONE-D-23-06410R1

Dear Dr. Mukwevho

We’re pleased to inform you that your manuscript has been judged scientifically suitable for publication and will be formally accepted for publication once it meets all outstanding technical requirements.

Kind regards,

Tunira Bhadauria, Ph.D.

Academic Editor

PLOS ONE

Additional Editor Comments (optional):

Reviewers' comments:

<quillbot-extension-portal></quillbot-extension-portal>

---

## [Editor Report · Acceptance letter]

22 May 2023

PONE-D-23-06410R1 

Long-term mammal herbivory on arthropod assemblages at Kruger National Park, South Africa 

Dear Dr. Mukwevho:

I'm pleased to inform you that your manuscript has been deemed suitable for publication in PLOS ONE. Congratulations! Your manuscript is now with our production department. 

Kind regards, 

on behalf of

Dr. Tunira Bhadauria 

Academic Editor

PLOS ONE